# Internal consistency of the IAGOS ozone and carbon monoxide measurements for the last 25 years.

**Romain Blot** [1], **Philippe Nedelec** [1], **Damien Boulanger** [2], **Pawel Wolff** [1], **Bastien Sauvage** [1], **Jean-Marc Cousin** [1], **Gilles Athier** [1], **Andreas Zahn** [3], **Florian Obersteiner** [3], **Dieter Scharffe** [4], **Hervé Petetin** [5], **Yasmine Bennouna** [1], **Hannah Clark** [6], **and Valérie Thouret** [1]

[1]Laboratoire d'aérologie (LAERO), CNRS UMR-5560 et Observatoire Midi-Pyrénées (OMP), Université de Toulouse, France
[2]Observatoire Midi-Pyrénées (OMP), Université de Toulouse, France
[3]Karlsruhe Institute of Technology (KIT), Institute for Meteorology and Climate Research (IMK), Karksruhe, Germany
[4]Max-Planck-Institut für Chemie (MPI), Air Chemistry Division, Hahn-Meitner-Weg, 55128 Mainz, Germany
[5]Barcelona Supercomputing Center (BSC), Barcelona, Spain
[6]IAGOS-AISBL, 98 Rue du Trone, Brussels, Belgium

**Correspondence:** R.Blot (romain.blot@aero.obs-mip.fr)

## Abstract.

The In-service Aircraft for a Global Observing System is a European research infrastructure that equips Airbus A340/330 with a system for monitoring atmospheric composition. The IAGOS instruments have three different configurations: IAGOS-CORE, IAGOS-MOZAIC and IAGOS-CARIBIC. Since 1994, there have been a total of 17 aircraft equipped. In this study, we perform an inter-comparison of about 8000 landing and take-off profiles to compare the O3 and CO measurements performed from these different configurations. The collocated profiles used in the study met various selection criteria. The first was a maximal 1 hour time difference between an ascent or descent by two different aircraft at the same airport and the second was a selection based on the similarity of air masses based on the meteorological data acquired by the aircraft. We provide here an evaluation of the internal consistency of the O3 and CO measurements since 1994. For both O3 and CO, we find no drift in the bias amongst the different instrument units (6 O$_3$ and 6 CO IAGOS-MOZAIC instruments, 9 IAGOS-CORE Package 1 and the 2 instruments used in the IAGOS-CARIBIC aircraft). This results gives us confidence that the entire IAGOS data base can be treated as one continuous program, and is therefore appropriate for studies of long-term trends.

## 1 Introduction

The In-service Aircraft for a Global Observing System (IAGOS, https://www.iagos.org) is a European research infrastructure (Petzold et al., 2015) that was officially launched in 2011 to equip Airbus A340/330 long-haul passenger aircraft with a newly designed system (named IAGOS-CORE) for collecting data on gases, aerosol and trace species throughout the tropopshere and lower stratosphere, and by maintaining the fleet of the former projects Measurement of OZone and water vapor by Airbus In-service AirCraft (MOZAIC now IAGOS-MOZAIC; Marenco et al. (1998)) and the Civil Aircraft for the Regular Investigation of the atmosphere Based on an Instrument Container (CARIBIC I & II now IAGOS-CARIBIC; Brenninkmeijer et al. (1999, 2007), https://www.caribic-atmospheric.com).

The IAGOS program was inspired by the The Global Atmospheric Sampling Program (GASP) started by the National Aeronautics and Space Administration (NASA) in 1975 (Perkins and Papathakos, 1977; Falconer et al., 1978; Falconer and Pratt, 1979) that showed that civil aircraft can serve as a new kind of observing platform for the atmosphere and can provide high temporal and spatial resolution for a relatively low cost compared with dedicated research aircraft field campaigns (Eyres and Reid, 2014). In the 1990s, the European aircraft manufacturer Airbus, concerned by the probable growing impact of the aeronautical industry on climate, has supported the Centre National de la Recherche Scientifique (CNRS) to develop the program MOZAIC with 5

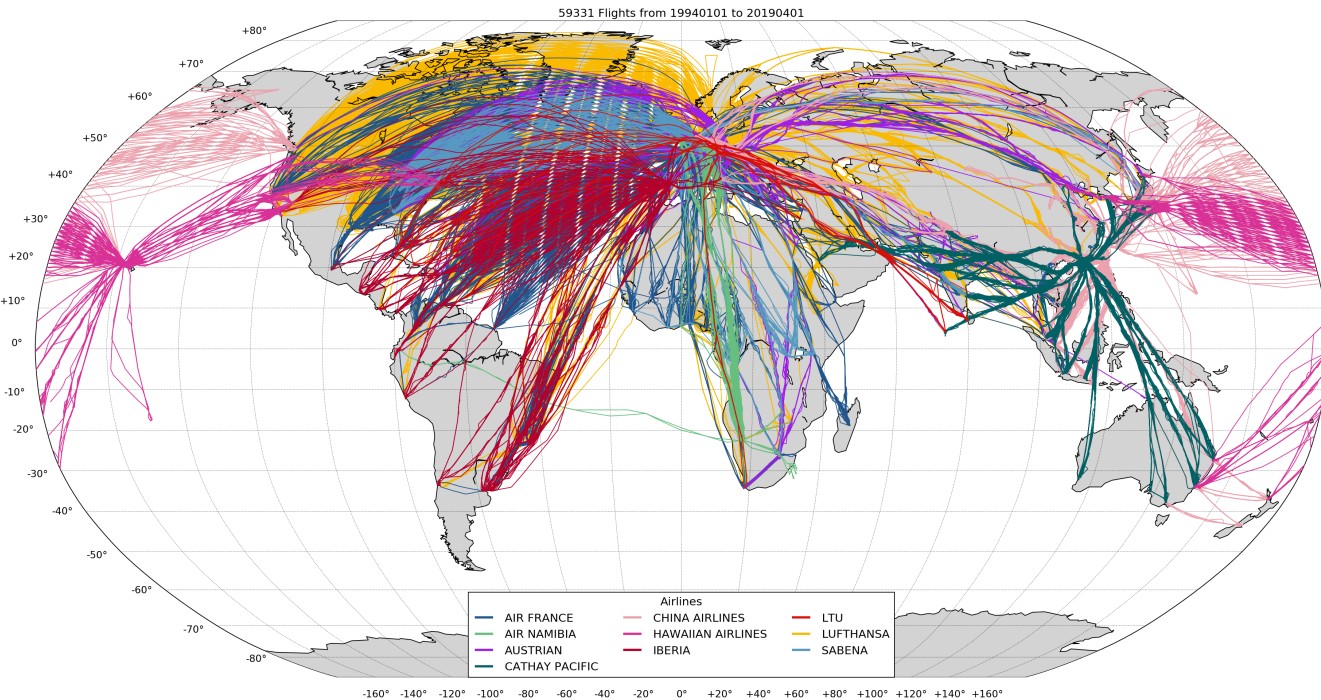

**Figure 1.** IAGOS aircraft fleet routes per airlines since 1994.

long-range Airbus A340s permanently equipped with sensors to sample O$_3$ (Thouret et al., 1998), water vapor (Helten et al., 1999; Smit et al., 2014), CO (since 2001; Nedelec et al. (2003)) and NOy (2000-2005; Volz-Thomas et al. (2005)). At about the same time, the program CARIBIC was launched in Germany with a different but complementary approach compared with MOZAIC. In MOZAIC, measurements of few key atmospheric components are taken on every flight made by the aircraft. In CARIBIC, every month, a one-ton-capacity aircraft freight container is loaded onto an aircraft to sample a large number of atmospheric species (up to 100 species). Originally planned for few years of operation, strong long term support from the French and German ministries of research, the European union (UE Framework Program fundings) and the participating airlines (Air France, Lufthansa, Iberia, Cathay Pacific, China Airlines, Hawaiian Airlines, Austrian Airlines, Air Namibia, Sabena) has allowed IAGOS to equip 17 Aircraft in 25 years (9 retired), delivering an impressive amount of measurements, both at cruising altitude (about 180 hpa) and during landings and take-offs over cities. It represents more than 60000 flights of 6 to 8h duration in average and around 120000 profiles over 338 cities. In total, it is about 3.5M points of observation spread around the world (see Figure 1). A more detailed overview of the IAGOS program and instruments can be found in Petzold et al. (2015).

Here we focus on the O$_3$ and CO data measured within the IAGOS program. These two atmospheric components have been identified as Essential Climate Variables (GCOS, 2010)

for which long-term monitoring is a key requirements for climate change projections. O$_3$ is the third most effective climate forcer in the Upper Troposphere/Lower Stratosphere (UTLS) after CO2 and CH4 (IPCC, 2013) and it has a detrimental impact on the human health. CO leads the production of O$_3$ by oxidation by the hydroxyl radicals (OH) and, at the same time, affects the oxidation potential of the troposphere (CO can act to both create and destroy OH) (Feilberg et al., 2002). CO, as primary pollutant formed by combustion processes is also a good proxy to track troposphere-stratosphere vertical transport and transcontinental transport pathways of plumes due to its relative long life time.

Until the 1990s, ozonesondes (i.e. the World Ozone and Ultraviolet Radiation Data Centre (WOUDC)) used to be the primary source of information on long-term changes of tropospheric O$_3$ (Tanimoto et al., 2015) besides the monitoring ground stations (i.e. the World Data Centre for Greenhouse Gases (WDCGG)) for the lower troposphere. For CO, the global sustainable observations for the troposphere came in 1999 with the satellite MOPITT (Measurements of Pollution in the Troposphere), also supported for validation by ground stations and research aircraft campaigns. O$_3$ and CO measurements have been part of the MOZAIC system since 1994 and 2001, respectively, with a large spatial and temporal coverage over different regions. Compared with other platforms, the IAGOS measurements are in-situ and the sampling techniques and the calibration strategies have remained the same since the beginning of the program (Nédélec et al., 2015). The avionic setup and the certification processes have

evolved to comply with current aeronautical civil safety regulations. Because of this long-term continuity, the IAGOS dataset is particularly adapted to studies of decadal trends and climatologies (Thouret et al., 2006; Zbinden et al., 2006; Hess and Zbinden, 2013; Zhang et al., 2016; Petetin et al., 2016b,a, 2018b; Gaudel et al., 2018; Cohen et al., 2018) and also air quality studies in and around urban agglomerations in the lower troposphere (Petetin et al., 2018a) thanks to the many international airports that serve big cities.

Supported by the European project IGAS (see Petzold et al. (2015) and http://www.igas-project.org/) launched in 2013, great efforts have been made to document standard operating procedures and to implement robust Quality Assurance and Quality Control (QA/QC) procedures for each measured atmospheric component. As a long term monitoring program and planned to last as long as it gets support from the aeronautical industry and public research entities, it is of primary importance to guarantee the traceability of the data and to regularly track the performances of the instruments in order to ensure a consistent time-series. In this paper, we present results which are part of the procedures for the QA/QC routinely performed within the IAGOS program. We investigate the internal consistency of the IAGOS dataset over the period 1994-2020, for O$_3$ and CO, by inter-comparing colocated IAGOS measurements obtained from the different aircraft of the fleet.

This study is organized as follows: section 2 describes the the instrumentation. The focus here is not to provide details on the IAGOS-MOZAIC, IAGOS-CORE and IAGOS-CARIBIC aeronautical system setup since all the descriptions can already be found in their associated publications Marenco et al. (1998), Nédélec et al. (2015) and Brenninkmeijer et al. (2007). Similarly, description of the corresponding instruments measuring O$_3$ and CO can also be found in Thouret et al. (1998), Nedelec et al. (2003), Nédélec et al. (2015), Zahn et al. (2012) and Scharffe et al. (2012). However, for a smooth reading of the paper, some key details of the project concepts are provided. Each sub program will hereafter be referred to as MOZAIC, IAGOS and CARIBIC (all being part of the IAGOS infrastructure). In the section 3, we will briefly describe the Quality Assurance (QA) part of the Standard Operation Procedures (SOPs) that each O$_3$ and CO sampling units undergoes before, during and after installation on the different IAGOS aircraft. In the two last sections, the methodology used to evaluate long-term internal consistency IAGOS O$_3$ and CO and then the global results since 1994 are presented.

## 2 Instrumentation

### 2.1 Concept overview

Equipping passenger aircraft with scientific instrumentation for atmospheric observations requires a unique and original approach in order to match scientific needs with the safety rules in the airline industry. Airborne observation programs using civil aircraft as a measurement platform share the same core characteristics. First, in-flight operations (system powering, measurements, calibration, data acquisition and safety checks) must be completely automatic with no attention required by the flight crew. Secondly, system maintenance should never interfere with the aircraft schedules. Finally, and maybe the most challenging, all equipment and structural modifications added to the aircraft (support racks, inlets, etc ...) must meet the requirements of either European Aviation Safety Agency (EASA), Federal Aviation Administration (FAA) or any other legal airworthiness institution. This is covered by the deliverance of a supplemental type certificate (STC). All measuring, controlling and safety systems, are powered by the aircraft facilities. Besides the deployment of dedicated scientific instruments, flight navigation and meteorological (see Table 1) data made by the aircraft system itself are collected using the Aeronautical Radio (ARINC) Inc. protocol.

**Table 1.** Parameters provided by the A340/A330 aircraft system.

| Name | Unit |
| --- | --- |
| Barometric altitude [a] | m |
| Radio altitude | m |
| GPS altitude | m |
| Latitude/Longitude | degrees |
| Meridional wind speed | ms$^{-1}$ |
| Zonal wind speed | ms$^{-1}$ |
| vertical wind speed | ms$^{-1}$ |
| Altitude rate | ms$^{-1}$ |
| Wind speed | ms$^{-1}$ |
| Wind direction | $^\circ$ |
| Aircraft ground speed | ms$^{-1}$ |
| Aircraft air speed | ms$^{-1}$ |
| Mach number | |
| Total air pressure | hPa |
| Left Static Pressure | hPa |
| Right Static Pressure | hPa |
| Total air temperature | $^\circ$Celsius |
| Static air temperature | $^\circ$Celsius |
| Track angle | degrees |
| Roll angle | degrees |
| Pitch angle | degrees |
| True heading | degrees |
| Track angle | degrees |

[a] Above mean sea level

**Table 2.** List of the aircraft Manufacturer Serial Number (MSN) equipped with the IAGOS-MOZAIC, IAGOS-CORE and IAGOS-CARIBIC system since 1994.

| Project | n° | Airline | Type | MSN | Tail Sign | Installed in : | Based in : | Dates |
|---|---|---|---|---|---|---|---|---|
| IAGOS-MOZAIC | 1 | Air France (AFR) | A340 | 49 | retired | Airbus Toulouse | Paris | 1994 to 2004 |
| | 2 | Multiple Airlines [b] | A340 | 51 | retired | Airbus Toulouse | Paris/Frankfurt Bruxelles/Windhoek | 1994 to 2013 |
| | 3 | Lufthansa (DLH) | A340 | 35 | retired | Airbus Toulouse | Frankfurt | 1994 to 2014 |
| | 4 | Lufthansa (DLH) | A340 | 53 | retired | Airbus Toulouse | Frankfurt | 1994 to 2014 |
| | 5 | Austrian (AUS) | A340 | 75 | retired | Airbus Toulouse | Vienna | 1995 to 2006 |
| IAGOS-CORE | 6 | Lufthansa (DLH) | A340 | 304 | D-AIGT | Hamburg/Frankfurt | Frankfurt | 2011 to now |
| | 7 | China Airlines (CAL) | A340 | 433 | retired | Taipei | Taipei | 2012 to 2017 |
| | 8 | Air France (AFR) | A340 | 377 | retired | Paris | Paris | 2013 to 2019 |
| | 9 | Cathay Pacific (CPA) | A330 | 421 | B-HLR | Xiamen | Hong-Kong | 2013 to now |
| | 10 | Iberia (IBE) | A340 | 221 | retired | Tel-Aviv/Madrid | Iberia | 2014 to 2016 |
| | 11 | Lufthansa (DLH) | A330 | 989 | D-AIKO | Malta | | 2015 to now |
| | 12 | China Airlines (CAL) | A330 | 861 | B-18317 | Taipei | Taipei | 2016 to now |
| | 13 | Hawaiian Airlines (HAL) | A330 | 1259 | N384HA | Brisbane/Honolulu | Honolulu | 2017 to now |
| | 14 | Air France (AFR) | A330 | 657 | F-GZCO | Xiamen | Paris | 2017 to now |
| | 15 | China Airlines (CAL) | A330 | 838 | B-18316 | Taipei | Taipei | 2017 to now |
| IAGOS CARIBIC I | 1 | LTU | Boeing 767 | 24259 | retired | Hamburg | Munich | 1997 to 2002 |
| IAGOS CARIBIC II | 2 | Lufthansa (DLH) | A340 | 540 | D-AIHE | Hamburg | Munich | 2004 to 2020 |

[b] This aircraft was operated first by Air France then Sabena (ex-Sogerma) (SAB) then Lufthansa and finally by Air Namibia (SW)

## 2.2 Fleet

Table 2 presents the list of aircraft that have been equipped with the MOZAIC, the IAGOS and CARIBIC systems. Since 1994, 9 international airlines, with in total 17 aircraft, with their home bases in various airports, have joined the programs. For MOZAIC, 5 Airbus A340 were equipped before delivery during the aircraft manufacturing at the Airbus facilities in Toulouse. This process was not applied for the IAGOS system. It can be noted that aircraft with Manufacturer Serial Number (MSN) lower than 100 carried the MOZAIC measuring system for over 10 to 20 years until the last aircraft retired in 2014. The CARIBIC cargo container first flew on a Boeing 767 from LTU between 1997 and 2002 before an A340 (D-AIHE) from Lufthansa (DLH). First the DLH IAGOS aircraft, the A340 MSN 304 D-AIGT, joined the fleet in 2011. During the following 6 years, 9 additional A340s or A330s were equipped with the system. For IAGOS, the aircraft modification is performed during the long maintenance lay-overs which occur roughly every 5 to 10 years of aircraft

lifetime. Some IAGOS equipped aircraft are retired earlier than others depending on the requirements of the airline. Between 2011 and 2014, 4 IAGOS, 1 CARIBIC aircraft and 3 MOZAIC A340s were operating. This allow us, later in this study, to compare the performance of the three different systems together.

## 2.3 System setups

Figure 2 shows the MOZAIC system installation inside one of the 5 equipped A340s. All the mechanical parts and other equipments were covered by an Airbus certification. An instrument cabinet rack was located at the starboard side in the avionics bay below the cockpit. The rack was composed of five shelves that received removable/replaceable units. The upper shelf contained a commercial $O_3$ analyzer (Thermoscientific, Model 49), the computer that controls the safety of the systems, the system start at takeoff and the stop at landing, the ARINC data acquisition and the data backups. On the third shelf, there was a modified $CO$ instrument (Thermoscientific, Model 48 CTL, Nedelec et al. (2003)) (this is

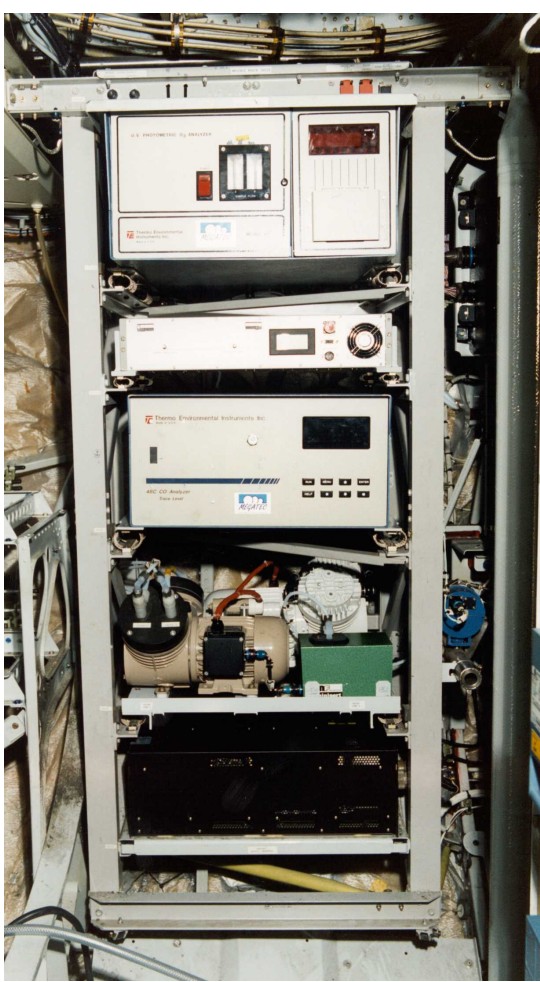

**Figure 2.** Picture of the MOZAIC system located on a A340 starboard side. See text for a brief description.

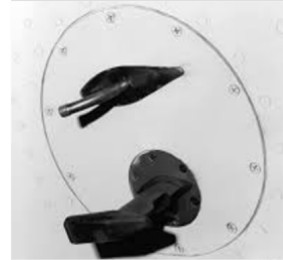
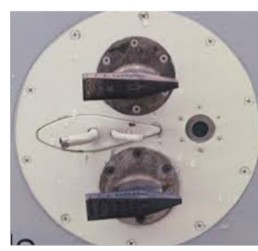

(a) MOZAIC (1994-2014)    (b) MOZAIC (D-AIGI : 2001-2014)

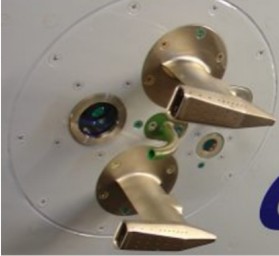
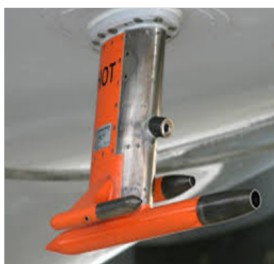

(c) IAGOS (since 2011)    (d) CARIBIC II

**Figure 3.** Pictures of the inlet plates used throughout the various programs. In a), b) and c), air for $O_3$ and $CO$ measurements is collected through the small facing forward pitot tube. In c), it is done using the lowest small inlet. Other inlets and Rosemount pitots are for water, aerosol or other gases measurements. The inlet shown in b) was installed on 1 MOZAIC aircraft (D-AIGI) with an additional Rosemount inlet for $NO_y/NO_x$ measurements. It is the precursor of the IAGOS inlet.

the configuration post-2001), and below, the pressurization pumps (1 for $CO$, 1 for $O_3$) that drive the air from outside the aircraft through an inlet plate (see Figure 3a and 3b) located on the fuselage at the port side. The data were stored on PCMCIA disks replaced roughly every 2 months. In case of $O_3$ and $CO$ instrument failure, the units could be independently replaced by spares. In total, through the MOZAIC period, from 1994 to 2014, 6 identical $O_3$ analyzers and 6 identical $CO$ analyzers with the same measurement uncertainties (see Table 3) were dispatched over the 5 MOZAIC aircraft, meaning that during the deployment period, only 1 spare was available at any one time. In the rest of this study, MOZAIC instrument Serial Numbers (SN) are referred to $SN_{PM}$ (ex: $01_{PM}$, $02_{PM}$, $03_{PM}$, etc ...).

In 2009, CNRS and Forschungszentrum Jülich (FZJ) initiated the project to modernize the MOZAIC system to produce a more sustainable concept that would be compliant with the safety regulations of any country. Figure 4 shows a picture of the system inside an A330 aircraft (which shares

a similar fuselage to the A340). The setup differs totally from the MOZAIC system. The new cabinet rack is located in the avionics compartment on the aircraft's port side, close to the inlet plate. One of the reasons for changing the position of the cabinet rack from the starboard side to the port side is that on modern aircraft the area that used to house the MOZAIC cabinet is usually occupied by the in-flight entertainment system. The advantage of the new position on the port side, was that it is closer to the inlet plate. The cabinet rack was completely redefined in order to house 3 removable boxes: One for the pressurization pumps, one for $O_3$ and $CO$ measurements, the so-called "Package 1" (or P1), and a third box for either one of the 2 optional certified "Package 2" (or P2), one for CO2 and CH4 measurements (Filges et al., 2015) and one for NOx measurements (Berkes et al., 2018). Compared with MOZAIC for which $O_3$ and $CO$ were acquired by 2 separate instruments, the choice for IAGOS was to compact both units into the same box. The measurement characteristics , however, remain the same (see Table 3). P1 also serves as central data acquisition system that collect the aircraft ARINC data, the IAGOS Capacitive Hygrometer (ICH, Helten et al. (1999)) data, data from the Backscatter Cloud Probe (BCP, (Beswick et al., 2013)), and data from the Package 2 if installed. The data files are transfered to the IAGOS server at

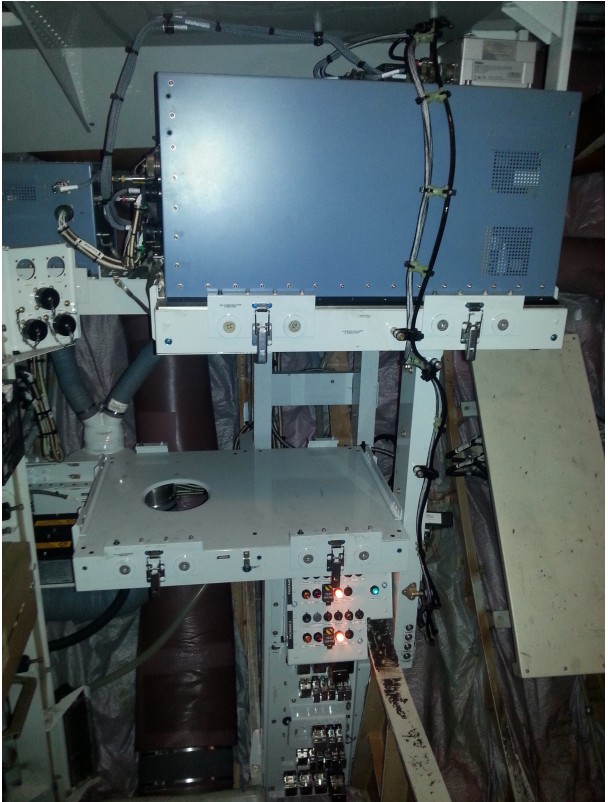

**Figure 4.** Picture of the IAGOS system setup in avionics bay. The Pump Box (small blue box on the upper corner left) embeds the pressurization pumps that drive the outside air collected through the pitot (see Figure 3c) to the Package 1 (P1, big blue box). The system functioning and safety is done via the control panel and relay panel seen on the lower part of the picture. The empty shelf is available for Package 2 instruments also developed within the IAGOS program.

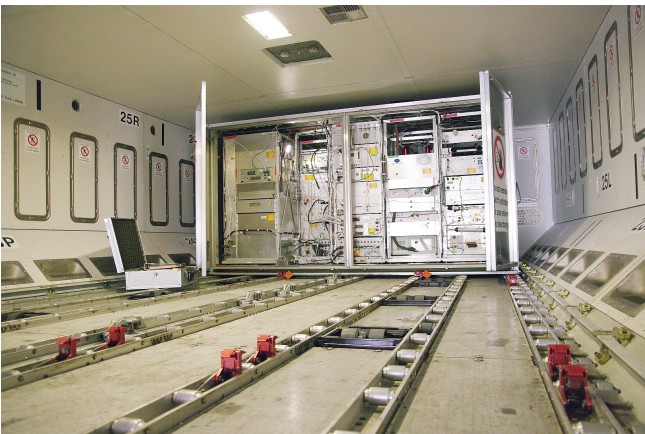

**Figure 5.** Picture of the CARIBIC laboratory container (1.65 tons, 19 instruments in 2017) inside the aircraft cargo bay.

CNRS in Toulouse by GSM each time the aircraft cargo door is opened/locked-up on ground and if the modem manages to connect to 2G/3G network for several minutes. In 2020, 14 P1 units were distributed over 7 IAGOS aircraft. There-
5 fore, there was 1 spare unit per aircraft. Each P1 has a serial number from 02 to 16. Serial number 01 was a qualification prototype that has not been used since the operational phase of the program started in 2011, and there is no serial number 13.
Within the CARIBIC I & II programs, $CO$ and $O_3$ measurements were performed by two separate units embedded into an modified air freight cargo container that additionally contained more than a dozen other instruments with a total weight of about 1.5 tons (Figure 5). The Lufthansa aircraft,
which host the CARIBIC container, were modified for the structural addition of an certified inlet system that holds different air intake probes (see Figure 3d) whether for aerosols, trace gases or water sampling. The container is loaded onto the aircraft for sequences of 4 to 6 flights with variable des-
tinations. After take-off, the main power supply is switched

on and a master computer (also in the container) takes command of all instrument activations/deactivations in addition to the acquisition of the ARINC parameters (see tables 1) and other data concerning the functional status of the container. In contrast to MOZAIC and IAGOS which start the trace 25 gases measurements during the take-off phase (aircraft speed $> 25\mathrm{ms}^{-1}$), the master computer starts the measurements at cruise altitude for CARIBIC I and when the barometric altitude (baro-altitude) is higher than about 2.5km above ground for CARIBIC II. For that reason and because this study will 30 focus only on profiles, the data from CARIBIC I will be discarded. $O_3$ measurements are performed by an instrument custom-made by Karlsruher Institut für Technologie (KIT) that combines the use of two sampling techniques in one box; the chemiluminescence of a dye in reaction with $O_3$ and 35 the absorption of the UV light by $O_3$. CO mixing ratios are provided, in CARIBIC II, by a custom-improved version of the Aero-Laser Model AL 5002 fast-response UV resonance fluorescence instrument. Characteristics, precisions and uncertainties are also summarized in table 3 for both instru- 40 ments. CARIBIC aircraft flight sequences are roughly every 30 to 60 days (depending on routes, aircraft availability and the availability of the instruments), therefore the need to have several spare units is less crucial compared with IAGOS and MOZAIC. Between each period of operation, all instru- 45 ments can be maintained and redeployed for the next flight sequences.

## 3   Standard operation procedures

### 3.1   MOZAIC and IAGOS

Before deployment, each measuring unit is cleaned and 50 maintained by a subcontractor that holds an EASA Part 145

**Table 3.** Summary of the CO and $O_3$ instrument characteristics. Maximum spatial resolutions are just indicative values and are given for civil aircraft approximated maximum ascent and descent speed of $15\mathrm{ms}^{-1}$ and $250\mathrm{ms}^{-1}$ during cruise.

| Programs | Techniques | | Detection limit; Uncertainties (accuracy; precision; integration time or frequency (f)) | | Maximum spatial resolutions (vertical ; horizontal) | |
|---|---|---|---|---|---|---|
| | $O_3$ | CO | $O_3$ | CO | $O_3$ | CO |
| IAGOS-MOZAIC & IAGOS-CORE | UV absorption | IR correlation | 2 ppbv; $\pm 2$ ppbv; $\pm 2\%$ ; 4 sec | 5 ppbv; $\pm 5$ ppbv ; $\pm 5\%$ ; 30s | 60m ; 1km | 450m ; 7.5km |
| IAGOS-CARIBIC | combination of 2 instruments : dry chemiluminescence + UV absorption | resonance fluorescence in the vacuum UV | 2ppbv; $\pm 2$ppbv;$\pm 2\%$; down to 10hz | 2ppbv;$\pm 1$ppbv for CO below 50ppbv, else $\pm 2\%$.; 1-2s | 1.5m; 25m | 15m; 250m |

agreement[1] and according to the corresponding Components Maintenance Manual (CMM), the latter being an official traceable document in regard of the awareness safety rules. All the performed tasks are also traced within an CNRS internal QA/QC document opened for each new maintained unit. After maintenance, $O_3$ and CO from P1 and the MOZAIC instruments are calibrated in the laboratory in Toulouse by the CNRS. For $O_3$, it is performed by comparison with a Thermo-Scientific Model 49PS reference instrument at several concentration levels to also check the instrument linearity within 1%. The $O_3$ reference is sent once a year to the French Laboratoire National d'Essais (LNE) for comparison with a traceable National Institute of Standards and Technology (NIST) instrument. For CO, we use a NIST referenced CO cylinders (CO in N2, 500 ppmv) and a calibrated dilution system. Calibration is performed for several levels of CO to control the linearity of the instrument within 2%. The CO dilution system is also controlled every year by the French LNE for flow calibration. The last important step in the deployment process is the systematic comparison with a MOZAIC measuring system (identical to Figure 2) that was kept in the laboratory to serve as a reference. Comparisons tests are performed, usually at night, using outside ambient atmospheric air, to make sure that the maintained units are robust and that the measurement difference with the reference instrument remained below 2% for $O_3$ and 5% for CO. After these tests have been made, the units are sent to the airlines for a scheduled installation within standard 6 month of operating time or for an unscheduled replacement if one instrument failed prematurely. The shipping logistics for all IAGOS parts are handled by the IAGOS Maintenance Center (IMC, Enviscope GmbH, http://www.enviscope.de).

The installation date of the unit (P1, PM O3 or PM CO) is reported in the QA/QC document as the start of Flight Period (FP) operation of the unit. The FP ends when the unit is removed from the aircraft. FPs do not depend on whether the instrument is performing measurements successfully and during this period all instrument failures, main aircraft events, maintenance actions by airline staff on the IAGOS system, and any noticeable issues that could impact the $O_3$ and CO measurements are reported and traced in the QA/QC document within this time. FPs are named using the aircraft MSN, the units SN and the number of the FP (ex: FP0989-10-P1SN04 for operations FP number 10 of P1 serial number 04 on DLH D-AIKO MSN0989). To refer to the aircraft, the SN is used instead of the tail sign since aircraft can be sold to an other airline during its operating life. Also, linked to the FP, instrument functional performances are reported and updated. This is usually done by flagging the functional parameters of the instrument according to previously defined thresholds of normally operating values.

After the six-month deployment on the aircraft, the instruments are returned to CNRS and the $O_3$ and CO instruments are calibrated and checked for drifts against the laboratory references. This is the last check before applying, if necessary, a correction to the data and to finally deliver the Level 2 (L2) data to the scientific community.

## 3.2 CARIBIC

Because the container is set up aboard the aircraft for only a few flights, the scientific instruments inside undergo less constraints due to the take off and landings. Therefore frequent laboratory-based maintenance are not necessary and these are performed roughly every two months (about 8 flights). The $O_3$ instrument maintenance is done in KIT and mostly consists in performing leak tests cleaning, pressure tests and the replacement of chemiluminescence sensor disc. All the maintenance tasks are traced filling out a maintenance list and shipping list before reintegration in the container. The functioning of the UV-photometer is controled each 4-6 months by comparison with a KIT custom-made laboratory $O_3$ instrument (using a Hg lamp as light source) and a long-path UV reference photometer (UMEG, GmbH) crosschecked by the World Meteorological Organization standard reference photometer n°15 at the Swiss Federal Laboratories

---

[1]Certification to the European Commission Regulation standards of design, production, maintenance and operation of aircraft components.

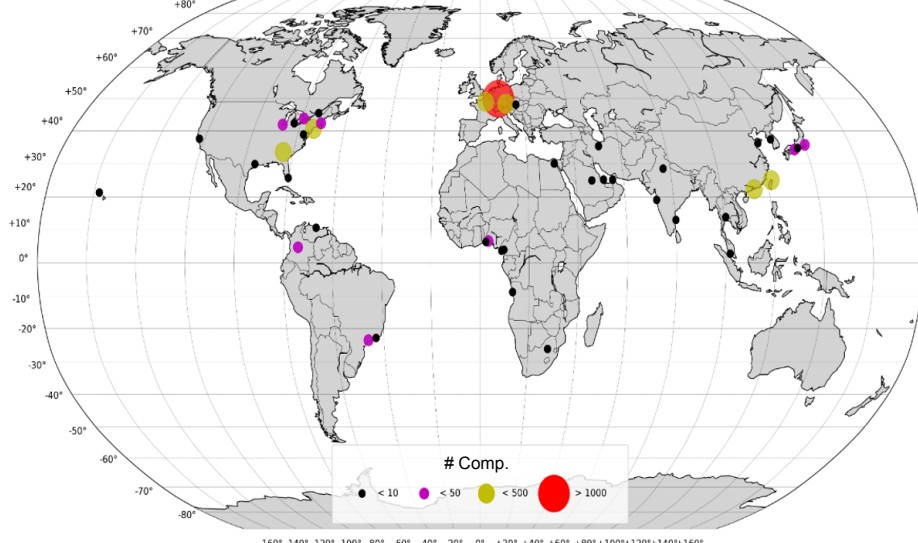

| | Airports | # Comp. |
|---|---|---|
| 1 | Frankfurt | 3068 |
| 2 | Taipei | 340 |
| 3 | Paris | 171 |
| 4 | Munich | 166 |
| 5 | New York | 121 |
| 6 | Hong Kong | 110 |
| 7 | Atlanta | 72 |
| 8 | Osaka | 45 |
| 9 | Tokyo | 29 |
| 10 | Chicago | 18 |
| 11 | Bogota | 14 |
| 12 | Toronto | 13 |
| 13 | Boston | 11 |
| 14 | Sao Paulo | 11 |
| 15 | Lagos | 11 |

**Figure 6.** World-map showing the total number of profile flight inter-comparisons observed over airport cities since 1994 for the entire aircraft fleet. The criteria for the maximal time coincidence is 1 hour ($\Delta t_{max} \leq 60$ min).

for Materials Science and Technology (EMPA) in Switzerland.

For the $CO$ instrument, the main important maintenance corresponds to the change of the $MgF_2$ $CO$ resonance lamp window every 3 years in order to maintain a high photon transmission statistic. This task was performed for the first time in June 2008. More details can be found in the associated publication Scharffe et al. (2012).

## 4 The internal consistency of IAGOS measurements

### 4.1 Background

The strategy of the program is to expand the number of aircraft equipped with the IAGOS system and to get more airlines (ideally, a minimum of 2 aircraft per Airline) involved in IAGOS. This has several advantages in addition to feed the atmospheric science community with data with more extensive global coverage. First, as maintenance actions can sometimes take weeks or months to be performed on a single aircraft, multiple aircraft are necessary to ensure the continuity of the time-series, which is particularly important for the studies of trends. Secondly, multiple aircraft offers the possibility to compare the different $O_3$ and $CO$ instruments that are installed on each aircraft by looking at trajectory coincidences in time and space. During cruise legs, trajectory coincidences (e.g. at least 2 aircraft that followed a quasi-identical route) are occasionally possible and are very useful in the data validation process. Unfortunately, they do not occur often enough to generate robust statistics. There are, however, many more landings or take-offs which fall within a maximum 1 hour ($\Delta t_{max} \leq 60$ min) time window and are

suitable for inter-comparison (see figure 6). Lufthansa has been deeply involved in IAGOS with several equipped aircraft since the beginning of MOZAIC, and with several aircraft equipped. As such, there are more than 3000 collocated profiles. In Taipei, there are 340 and in Paris 171. Airbus A330/A340 are long-haul aircraft which serve main international airports. The landing and take-off coincidences are not necessarily between aircraft of the same airline and more importantly, the mounted $O_3$ and $CO$ instruments are dispatched randomly. Consequently, thanks to the large number of daily coincidences, it is possible to perform a quality control on the $O_3$ and $CO$ measurements getting a fair idea on how each serial number instrument compared with the others according to some limitations inherent to the use of commercial aircraft, which are exposed in the following section.

### 4.2 Methodology

#### 4.2.1 MOZAIC and IAGOS

One of the main obvious limitations with the use of commercial aircraft as a scientific measuring platform is that air routes and departures/arrivals schedules are fixed by the airlines and, of course, strict airworthiness rules must be respected for the safety of the passengers. This implies explicitly that we can not have two aircraft flying too close from each other to perform proper flight inter-comparison exercises as it is often the case for field campaigns using multiple research aircraft. Therefore, even though as shown in the section above, there can be several IAGOS aircraft landing or taking off at the same airport within a time difference less than 1 hours (sometimes less than 10 minutes), we can not expect that they follow the same routes and, consequently,

that they always fly in the same physically and chemically steady air mass. It is also reasonable to not expect perfect 1:1 regression for the comparisons, however by choosing adequate screening criteria for air masses and flight track coincidences, it is possible to get a good estimate of the internal consistency of the instruments.

Figure 7 illustrates the method applied to each flight stored in the database at the CNRS server in Toulouse. We present the steps used for the operational phase of the QC procedure in the IAGOS program. All the data used in the following study are L2 final data provision (see also http://www.iagos-data.fr and Petzold et al. (2015) for details), which are the data that are distributed publicly. For each individual flight, a java script is triggered to look for any other flight in the database that has landed or taken off at the same airport within a time window of maximum 1 hour. Some examples of the testing phase of this procedure were presented in Nédélec et al. (2015). Figure 7 shows an example with one aircraft equipped with the MOZAIC system and an one aircraft equipped with IAGOS system that both took off at Frankfurt airport on 2012/12/17 with only a 13 minutes time difference. As it is often the case, the two aircraft quickly headed to different destinations taking different routes. After only few minutes, the distance between the two tracks can be several hundred kilometers. The profiles of O3 and CO measured by these two aircraft are plotted in Figure 7c. The curves do not refer to the name of the aircraft but to the serial number of the $O_3$ or $CO$ instrument for MOZAIC or the serial number for package 1 in the case of IAGOS. In this specific example, the MOZAIC aircraft is mounted with the $O_3$ instrument serial number $03_{PM}$ and $CO$ instrument serial number $04_{PM}$ that are compared with the IAGOS package 1 serial number 03. On these two figures, grey horizontal lines indicate where the air masses encountered by the two aircraft present similar characteristics considering the potential temperature (T), the wind direction (WindDir) and the potential vorticity (PV). The wind direction and the temperature are measured directly by the aircraft sensors (see Table 1). The atmospheric pressure and the absolute temperature are also measured directly by the aircraft and are used to derive the potential temperature. The potential vorticity (PV), which is often used to approximate the position of the dynamical tropopause that separates the upper troposphere from the stratosphere (Holton et al., 1995), is taken from ECMWF operational analyses and evaluated at the aircraft position (Sauvage et al., 2017) by the FLEXible PARTicle dispersion model (FLEXPART, Stohl et al. (2005)). Threshold values for the maximum differences at equal barometric altitude of these screening parameters are summarized in the table Figure 7b. To prevent the influence of highly variable mixing ratios due to local sources of pollution within the boundary layer, a lower baro-altitude limit is also set to 2 km.

Figure 7d shows the scattergram plot produced with the measurements made from the two aircraft and resulting from

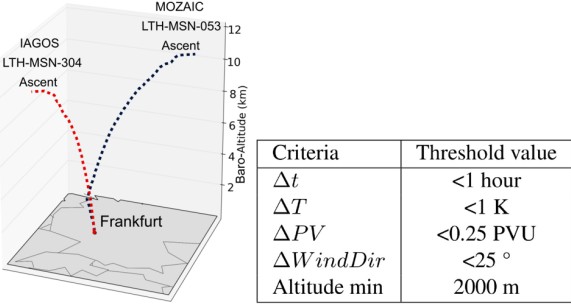

| Criteria | Threshold value |
|---|---|
| $\Delta t$ | <1 hour |
| $\Delta T$ | <1 K |
| $\Delta PV$ | <0.25 PVU |
| $\Delta WindDir$ | <25 ° |
| Altitude min | 2000 m |

(a) Aircraft ascent tracks.    (b) Air mass screening criteria.

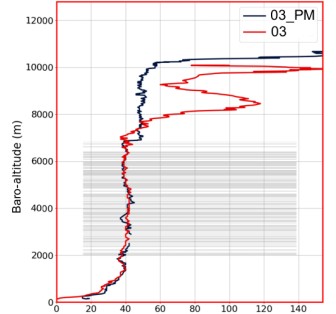 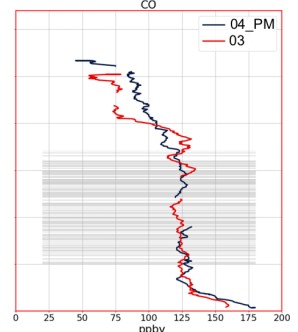

(c) $O_3$ (left) and $CO$ (right) measurement inter-comparison profiles between the Package MOZAIC (black) and IAGOS Package 1 (red) instruments. Grey lines correspond to the air masses matching according to the criteria in figure 7b. Instrument serial number are indicated in the legend.

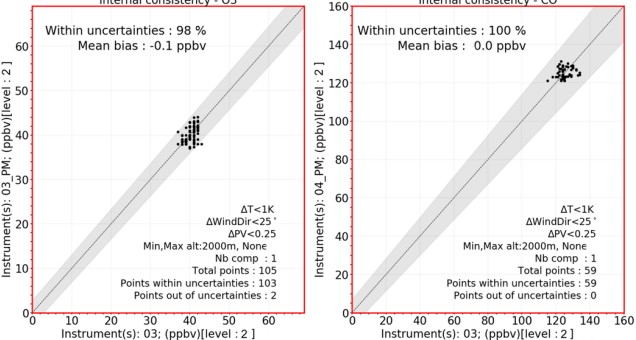

(d) $O_3$ (left) and $CO$ (right) scattergram plots comparing measurements from MOZAIC instrument (ordinate) and the IAGOS instrument (Abscissa). Dotted line is the 1:1 line, Grey area displays the total uncertainty for both instruments. Statistic information are also displayed for each components (see text).

**Figure 7.** $O_3$ and $CO$ profile inter-comparisons on 2012/12/17 between MOZAIC Luthansa msn-053 and IAGOS Luthansa msn-304. Both aircraft ascent from Frankfurt airport within a 13 minutes time interval. The MOZAIC aircraft is equipped with the instrument unit serial number $03_{PM}$ for $O_3$ and $04_{PM}$ for $CO$. The Package 1 mounted on the IAGOS aircraft is serial number 03.

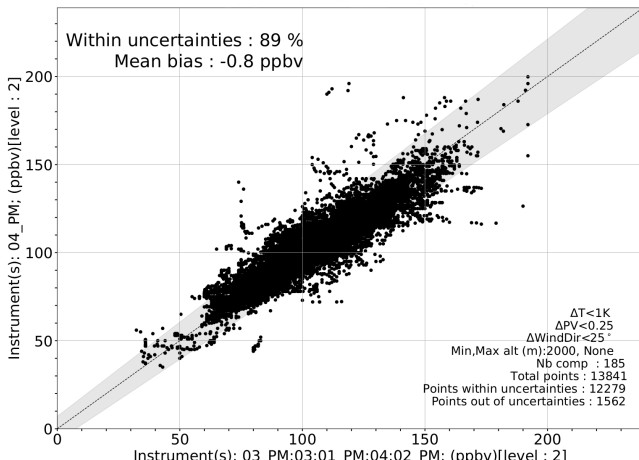

**Figure 8.** Scattergram plot compiling all the flight-intercomparisons found for CO MOZAIC instrument serial number $04_{PM}$ for the flight period number 13 on Lufthansa A340 msn53. The grey area represents the total measurement uncertainty. Statistic scores are presented in the figure legend.

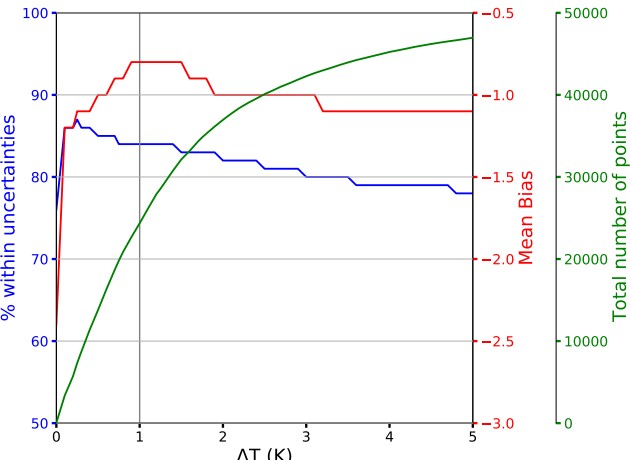

**Figure 9.** Percentage of points within the uncertainty and mean bias as function of the potential temperature difference measured by the co-located aircraft found for CO instrument $04_{PM}$ and for the flight period number 13 on Lufthansa A340msn53. The total number of comparable points according the air temperature difference is also shown. The maximum time difference between the landings or the take offs is 1 hour.

the air mass filtering process (the points highlighted by the grey horizontal lines in Figure 7c). The grey area represent the quadratic sum of the total uncertainties for each instruments. Since all O$_3$ and CO instruments for MOZAIC and IAGOS have the same characteristics, the grey zone, represents the area between $C \pm \sqrt{(2*\Delta C)^2}$ where C is the measured mixing ratio and $\Delta C$ is the total uncertainty of the measurements. These scattergram plots are produced routinely for each flight and at each validation step through the process of data validation. Besides plotting the 1:1 line and the associated uncertainty, 2 main key performance indicators are calculated. The first is the bias between the compared instruments. It is the mean of the distance from the 1:1 line for every points. The second is more an indication of the dispersion by calculating the percent of measurements that remain within the total instrument uncertainties. This is obtained if each compared measurement agrees with :

$$\frac{|\, C_{SN_{abscissa}} - C_{SN_{ordinates}}\,|}{\sqrt{(\Delta C_{SN_{abscissa}})^2 + (\Delta C_{SN_{ordinates}})^2}} \leq 1 \qquad (1)$$

As it can be noted, the inter-comparison plots chosen here, in Figure 7, correspond to an ideal case study. For O$_3$, instrument $03_{PM}$ measurements do not differ from P1 serial number 03 (mean bias almost zero) and 98% of the points are within measurement uncertainties. For CO, instrument $04_{PM}$ the result is even better.

However, the information that we really want to reach for the internal consistency is how each instrument performed globally through its entire flight period compared with other instruments flying during coincident periods. For example, CO MOZAIC instrument $04_{PM}$ of Figure 7 was installed on the Lufthansa A340 SN53 on December 2012 for a flight

period that last 448 days in total. It was the thirteenth time (FP n°13) that a CO instrument was operated on this aircraft. During this operating time, 185 flights were found within a 1 hour time window and 12279 points were found to be comparable according to the same air mass similarity criteria used to produce Figure 7. Figure 8 shows the scattergram plot that compiles all the available data for this flight period. On the abscissa, the serial numbers of all the CO instruments that were compared to $04_{PM}$ are highlighted. As it can be noted, the performance indicators present globally a very good score with a mean bias of -0.8 ppbv and 89% of the points within the total measurement uncertainties. Consequently, it can be stated confidently that the CO measurements of $04_{PM}$ on Lufthansa Airbus SN53 for FP n°13 present, on average, a negligible bias compared with the other instruments and in regard to the measurement uncertainties.

The air mass similarity criteria thresholds were found by testing the following method on several different FPs. First, as it is shown in Figure 9 for the CO instrument $04_{PM}$ (same FP than in Figure 8), we monitor the evolution of performance indicators as we increase the temperature difference threshold. The percentage of points within the uncertainty peaks at a temperature difference of 0.25K and the mean bias peaks at 1K. For this flight period, we found 185 co-located aircraft which explain the large number (about 10000) of comparable points, even for a very restrictive threshold. Seeing the rapid increase of the number of points and comparing the results from others instrument units, we found that a temperature difference threshold of 1K would be a better compromise for shorter FPs or for the ones with instruments

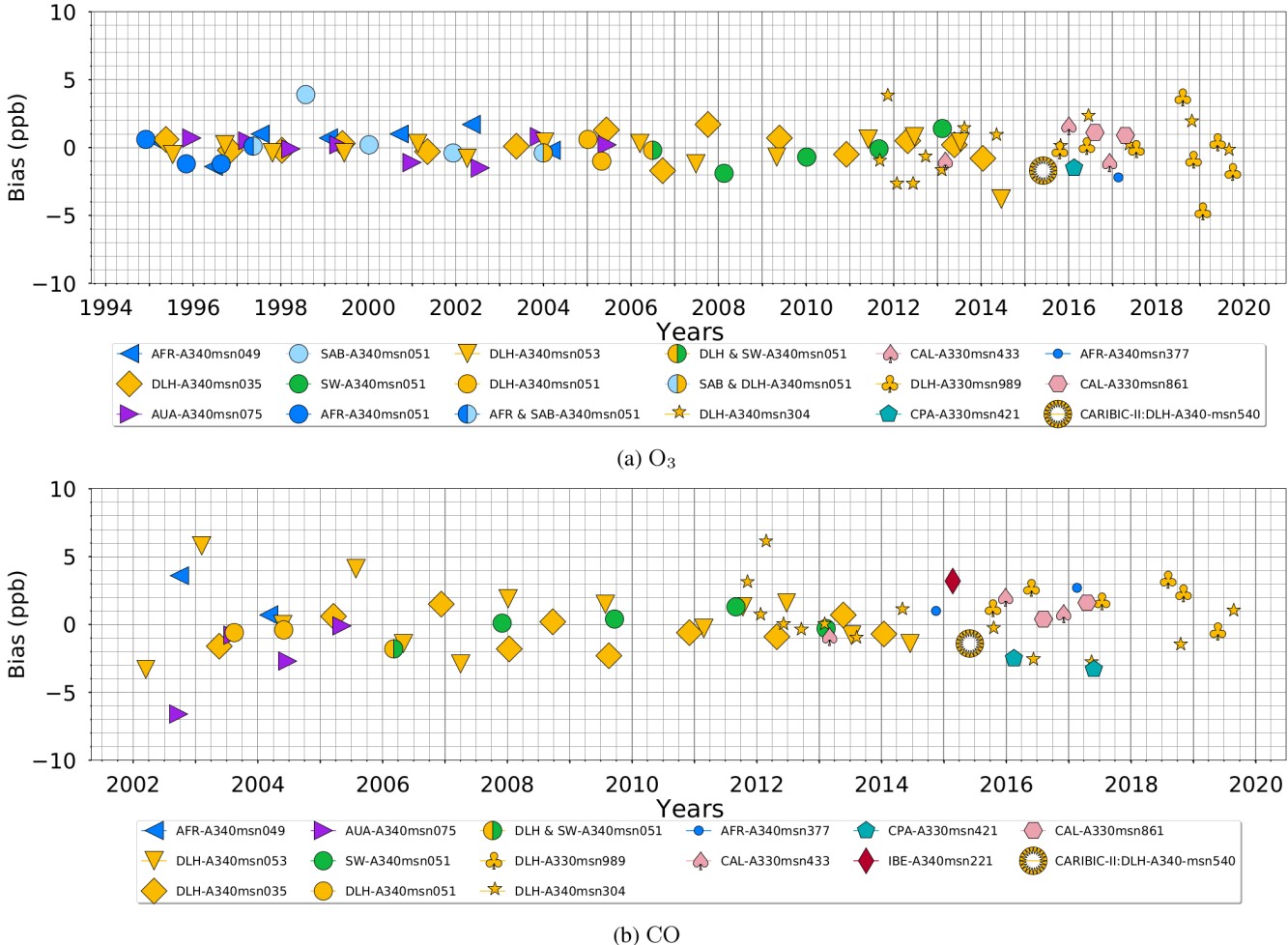

(a) $O_3$

(b) CO

**Figure 10.** Internal consistency of the long-term $O_3$ (a) and CO (b) measurement time-series within the IAGOS program. Different symbols are used to differentiate the aircraft and different colors are used for each airline. Symbol positions show the flight inter-comparison measurement mean bias for an intrument FP centered at the middle of the period. For CARIBIC, the symbol is position arbitrarily on the time scale (see text).

that are operated in remote area with less aircraft rotation (less co-located flights). To choose the two additional meteorological air mass similarity thresholds, we set the temperature thresholds to 1K and successively iterate on the wind direction and the potential vorticity difference increase for several FPs. Then, we decided to applied the same thresholds to all the FPs. For the time difference, we get better results for thresholds less than 1 hours (more steady meteorological conditions with respect to the life time of $O_3$ and CO), however, we found that, for example at Frankfurt airport, the number of co-located flights is reduced by 50% per 30 minutes.

This methodology is applied to each of the MOZAIC and IAGOS instruments that flew on the IAGOS fleet, for all flight periods since 1994 and for which data were delivered as L2 to the scientific community. The results are gathered in Table 4 and 5 and a summary study is exposed in section 5.

### 4.2.2 CARIBIC versus MOZAIC/IAGOS

The flight period clustering concept described above can not be applied to evaluate the performance of the CARIBIC measurements compare with those from IAGOS and MOZAIC because CARIBIC operates for only days every couple of months and therefore there are too few flight inter-comparisons per period (maximum 1 or 2 per flight sequences). However, if we apply the method considering the whole CARIBIC operation since 2005 until now, we found 101 and 114 flight inter-comparisons for $O_3$ and CO, respectively, with, in total, 7254 and 7286 points of comparisons and the vast majority of these profile coincidences where found over Frankfurt airport until 2014, period before DLH moved the aircraft to Dusseldorf as home airport. Frankfurt has remained the home base of all other MOZAIC

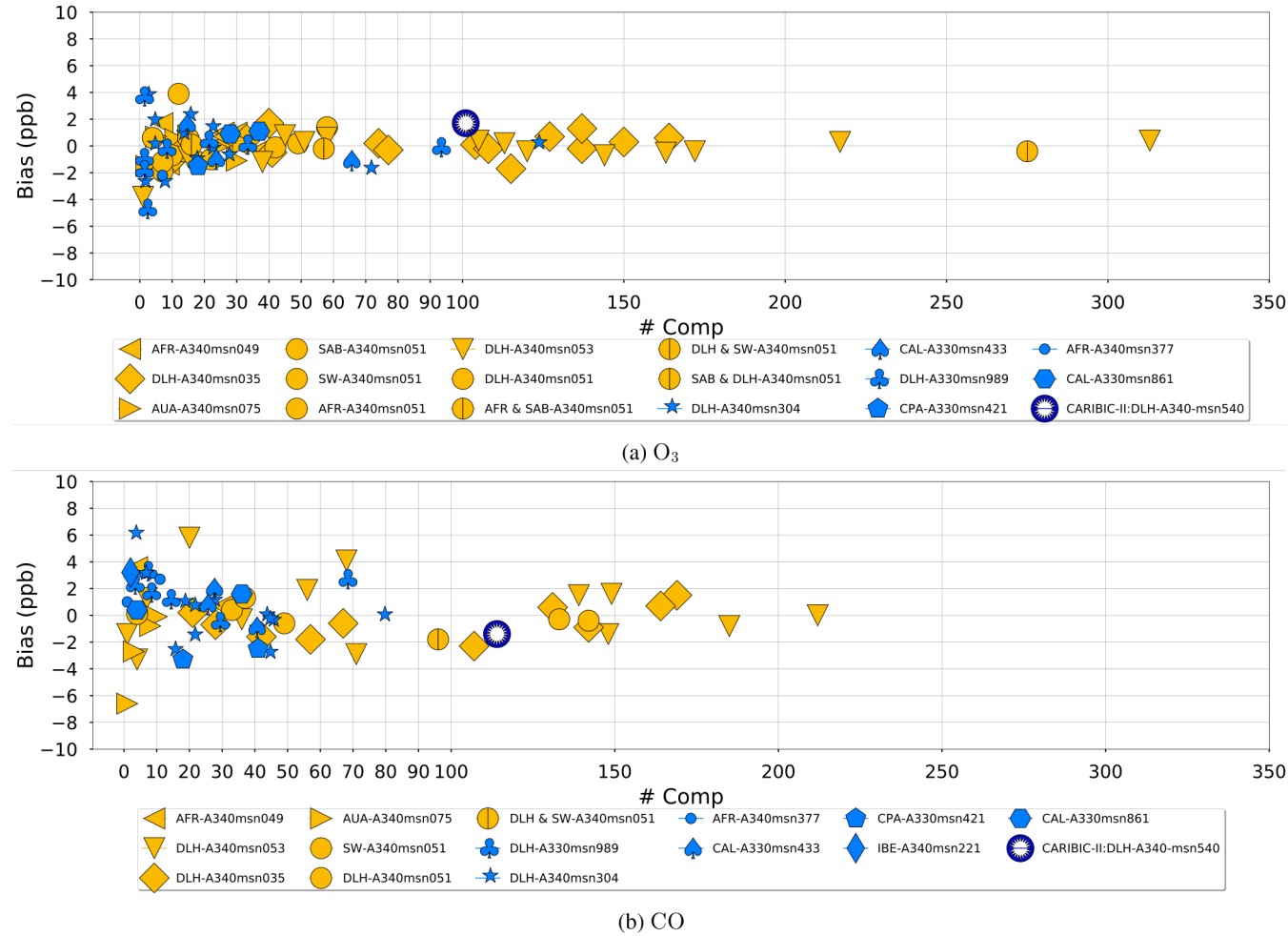

(a) $O_3$

(b) CO

**Figure 11.** Dependency of the $O_3$ (a) and CO (b) instrument measurement mean bias for each FP with the number of flight inter-comparison profiles. Different symbols are used for each distinct aircraft (same as Figure 10), the yellow color is for the MOZAIC program and the blue color is for IAGOS and CARIBIC.

and IAGOS aircraft. The performances indicators results for CARIBIC are also presented in section 5.

## 5 Results

The results presented in Figure 10 and Figure 11 consti-
5 tute the milestone of this study and more generally for the QA/QC process within the IAGOS program. It synthesizes the statistics gathered in Tables 4 and 5 and the outcomes of the methodology presented in section 4 applied to every $O_3$ and CO instrument that flew aboard every MOZAIC, IAGOS
and CARIBIC aircraft.

Figure 10 presents the time-series of the $O_3$ and CO mean Bias for all the instrument FPs from 1994 to 2020. For more clarity, the choice was made to cluster the results from each instrument's FP and for each aircraft's MSN. Different sym-
15 bols are assigned to distinguish different aircraft and a color is assigned for each different airline. Individual symbols rep-

resent the mean bias of each instrument compared with the others within their relative FP, excepted for CARIBIC (see section 4.2.2). For $O_3$, it is easily noticeable on Figure 10 that the large majority of the symbols (including CARIBIC) 20 remain within $\pm 2$ ppbv, which is the accuracy of the O3 Package 1 and Package MOZAIC instruments, and that symbols are homogeneously distributed around 0 ppbv through-out the entire IAGOS time-series since 1994. The same re-sults are evident for CO in Figure 10b with most of the sym- 25 bols falling within the accuracy of the CO instruments (i.e. +- 5ppbv), and the symbols are homogeneously distributed around 0 ppbv. Figure 11 differs from the previous one by showing the mean bias of the O3 and CO instruments ac-cording to the number of profiles that fit the criterium de- 30 scribed in section 4.2 for each instrument's FP since 1994. Each symbol is also associated with a distinct aircraft MSN, however only the colours yellow and blue are used in this Figure in order to differentiate the aircraft equipped with the

| Aircraft / MSN | Field | FP n°1 | FP n°2 | FP n°3 | FP n°4 | FP n°5 | FP n°6 | FP n°7 | FP n°8 | FP n°9 | FP n°10 | FP n°11 | FP n°12 | FP n°13 | FP n°14 | FP n°15 | FP n°16 | FP n°17 |
|---|---|---|---|---|---|---|---|---|---|---|---|---|---|---|---|---|---|---|
| MOZAIC Lufthansa MSN-035 | start date | 1994/07/31 | 1996/03/05 | 1997/07/09 | 1998/07/15 | 2000/04/12 | 2002/06/04 | 2004/05/05 | 2004/09/06 | 2006/03/12 | 2007/04/02 | 2008/04/08 | 2008/06/29 | 2008/09/18 | 2010/01/24 | 2011/07/10 | 2012/11/13 | 2013/11/26 |
| | instr. serial | 03_PM | 01_PM | 03_PM | 06_PM | 05_PM | 02_PM | 01_PM | 02_PM | 01_PM | 03_PM | 02_PM | 06_PM | 03_PM | 01_PM | 04_PM | 02_PM | 04_PM |
| | duration (days) | 583 | 491 | 371 | 637 | 783 | 700 | 125 | 552 | 386 | 371 | 82 | 82 | 493 | 624 | 400 | 378 | 98 |
| | mean bias | 0.6 | -0.2 | -0.2 | 0.3 | -0.3 | 0.1 | | 1.3 | -1.7 | 1.7 | | | 0.7 | -0.5 | 0.5 | 0.8 | -0.8 |
| | % consistency | 59 | 60 | 61 | 61 | 57 | 56 | | 62 | 59 | 53 | | | 69 | 54 | 68 | 68 | 63 |
| | # comparisons | 164 | 137 | 108 | 150 | 77 | 104 | | 137 | 115 | 40 | | | 127 | 54 | 33 | 74 | 11 |
| | # points | 27813 | 18167 | 14820 | 19126 | 10886 | 10733 | 0 | 17219 | 16759 | 5511 | 0 | 0 | 18650 | 4390 | 3051 | 7156 | 924 |
| MOZAIC Air France MSN-049 | start date | 1994/07/31 | 1995/12/19 | 1996/12/19 | 1998/02/04 | 2000/01/31 | 2001/04/29 | 2003/05/21 | | | | | | | | | | |
| | instr. serial | 01_PM | 06_PM | 04_PM | 05_PM | 03_PM | 01_PM | 04_PM | | | | | | | | | | |
| | duration (days) | 506 | 366 | 412 | 726 | 454 | 752 | 578 | | | | | | | | | | |
| | mean bias | 0.3 | -1.4 | 1.0 | 0.7 | 1.0 | 1.7 | -0.2 | | | | | | | | | | |
| | % consistency | 54 | 65 | 57 | 61 | 49 | 56 | 68 | | | | | | | | | | |
| | # comparisons | 26 | 9 | 26 | 61 | 30 | 7 | 38 | | | | | | | | | | |
| | # points | 3993 | 1327 | 3673 | 3555 | 5645 | 539 | 4556 | | | | | | | | | | |
| MOZAIC Multiple MSN-051 | start date | 1994/07/31 | 1995/04/04 | 1996/06/05 | 1996/11/06 | 1997/11/19 | 1999/04/05 | 2000/10/10 | 2003/02/05 | 2004/11/22 | 2005/02/20 | 2005/07/10 | 2007/06/20 | 2008/10/14 | 2008/12/15 | 2011/01/26 | 2012/04/05 | |
| | instr. serial | 03_PM | 02_PM | 04_PM | 05_PM | 02_PM | 01_PM | 04_PM | 06_PM | 03_PM | 06_PM | 05_PM | 01_PM | 02_PM | 06_PM | 03_PM | 01_PM | |
| | airline | Air France | Air France | Air France | AFR & SAB | Sabena | Sabena | Sabena | SAB & DLH | Lufthansa | Lufthansa | DLH & SW | Air Namibia | Air Namibia | Air Namibia | Air Namibia | Air Namibia | |
| | duration (days) | 247 | 428 | 154 | 378 | 502 | 554 | 848 | 656 | 90 | 140 | 710 | 481 | 63 | 772 | 435 | 621 | |
| | mean bias | 0.6 | -1.2 | -1.9 | 0.1 | 0.2 | 0.2 | -0.4 | -0.4 | 0.6 | -1.0 | -0.2 | -1.9 | | -0.7 | -0.1 | 1.4 | |
| | % consistency | 53 | 56 | 52 | 51 | 38 | 52 | 62 | 64 | 22 | 15 | 53 | 36 | | 49 | 57 | 51 | |
| | # comparisons | 15 | 18 | 7 | 16 | 12 | 49 | 13 | 275 | 4 | 22 | 57 | 7 | | 10 | 42 | 58 | |
| | # points | 2263 | 2926 | 1451 | 3091 | 1504 | 9295 | 2248 | 31634 | 36 | 970 | 6420 | 770 | 0 | 553 | 5255 | 5438 | |
| MOZAIC Lufthansa MSN-053 | start date | 1994/11/08 | 1996/03/19 | 1997/03/25 | 1998/05/24 | 2000/07/03 | 2001/10/04 | 2002/10/03 | 2005/04/21 | 2007/02/07 | 2007/11/12 | 2010/10/17 | 2012/01/18 | 2012/11/27 | 2014/02/18 | | | |
| | instr. serial | 04_PM | 03_PM | 06_PM | 04_PM | 06_PM | 03_PM | 05_PM | 03_PM | 02_PM | 01_PM | 02_PM | 06_PM | 03_PM | 06_PM | | | |
| | duration (days) | 497 | 371 | 425 | 771 | 458 | 364 | 931 | 637 | 278 | 1070 | 458 | 314 | 448 | 238 | | | |
| | mean bias | -0.5 | 0.2 | -0.4 | -0.4 | 0.3 | -0.8 | 0.4 | 0.3 | -1.2 | -0.7 | 0.6 | 0.8 | 0.4 | -3.8 | | | |
| | % consistency | 59 | 60 | 58 | 58 | 61 | 61 | 64 | 59 | 57 | 67 | 60 | 50 | 67 | 45 | | | |
| | # comparisons | 163 | 113 | 120 | 172 | 51 | 15 | 313 | 217 | 38 | 144 | 58 | 45 | 105 | 1 | | | |
| | # points | 27369 | 14707 | 16499 | 22267 | 7090 | 1506 | 34533 | 28240 | 6020 | 20617 | 6145 | 3619 | 9949 | 91 | | | |
| MOZAIC Austrian MSN-075 | start date | 1995/03/04 | 1996/10/06 | 1997/09/23 | 1998/09/24 | 1999/11/30 | 2002/01/29 | 2003/01/13 | 2004/10/12 | 2006/02/16 | 2006/05/12 | | | | | | | |
| | instr. serial | 05_PM | 02_PM | 01_PM | 03_PM | 02_PM | 06_PM | 03_PM | 01_PM | 06_PM | 02_PM | | | | | | | |
| | duration (days) | 582 | 352 | 366 | 432 | 791 | 349 | 638 | 491 | 85 | 171 | | | | | | | |
| | mean bias | 0.7 | 0.5 | -0.1 | 0.2 | -1.1 | -1.5 | 0.8 | 0.2 | | | | | | | | | |
| | % consistency | 53 | 35 | 72 | 51 | 49 | 60 | 63 | 57 | | | | | | | | | |
| | # comparisons | 5 | 11 | 24 | 32 | 30 | 2 | 15 | 13 | | | | | | | | | |
| | # points | 754 | 2019 | 3968 | 5388 | 5920 | 723 | 1632 | 626 | 0 | 0 | | | | | | | |
| IAGOS Lufthansa MSN-304 | start date | 2011/07/06 | 2011/10/04 | 2011/12/12 | 2012/02/08 | 2012/04/04 | 2012/07/12 | 2012/10/29 | 2013/05/29 | 2013/12/08 | 2014/09/02 | 2015/03/30 | 2016/04/13 | 2016/07/08 | 2018/02/22 | 2019/05/18 | | |
| | instr. serial | 03 | 03 | 03 | 03 | 03 | 03 | 03 | 03 | 03 | 04 | 03 | 04 | 03 | 04 | 09 | | |
| | duration (days) | 90 | 43 | 58 | 7 | 99 | 109 | 164 | 115 | 266 | 209 | 380 | 86 | 594 | 450 | 171 | | |
| | mean bias | -1.1 | 3.7 | -2.8 | | -2.8 | -0.8 | -1.8 | 1.3 | 0.8 | | 0.0 | 2.2 | 0.1 | 1.8 | -0.3 | | |
| | % consistency | 56 | 52 | 63 | | 49 | 47 | 65 | 53 | 63 | | 34 | 48 | 56 | 62 | 57 | | |
| | # comparisons | 17 | 2 | 7 | | 1 | 71 | 71 | 22 | 13 | | 4 | 15 | 123 | 4 | 22 | | |
| | # points | 821 | 672 | 380 | 0 | 57 | 2387 | 8202 | 1706 | 992 | 0 | 306 | 839 | 9617 | 467 | 2356 | | |
| IAGOS Air France MSN-377 | start date | 2013/06/14 | 2016/04/19 | | | | | | | | | | | | | | | |
| | instr. serial | 02 | 11 | | | | | | | | | | | | | | | |
| | duration (days) | 1040 | 609 | | | | | | | | | | | | | | | |
| | mean bias | | -2.2 | | | | | | | | | | | | | | | |
| | % consistency | | 53 | | | | | | | | | | | | | | | |
| | # comparisons | | 7 | | | | | | | | | | | | | | | |
| | # points | 0 | 1035 | | | | | | | | | | | | | | | |
| IAGOS Cathay Pacific MSN-421 | start date | 2013/08/01 | 2014/05/31 | 2015/05/13 | 2016/11/17 | | | | | | | | | | | | | |
| | instr. serial | 06 | 05 | 06 | 07 | | | | | | | | | | | | | |
| | duration (days) | 303 | 347 | 554 | 382 | | | | | | | | | | | | | |
| | mean bias | | | -1.5 | | | | | | | | | | | | | | |
| | % consistency | | | 32 | | | | | | | | | | | | | | |
| | # comparisons | | | 18 | | | | | | | | | | | | | | |
| | # points | 0 | 0 | 1618 | 0 | | | | | | | | | | | | | |
| IAGOS China Airlines MSN-433 | start date | 2012/06/21 | 2014/06/18 | 2014/12/17 | 2015/07/23 | 2016/05/10 | | | | | | | | | | | | |
| | instr. serial | 04 | 08 | 09 | 08 | 09 | | | | | | | | | | | | |
| | duration (days) | 485 | 182 | 218 | 292 | 888 | | | | | | | | | | | | |
| | mean bias | -1.0 | | | 1.6 | -1.1 | | | | | | | | | | | | |
| | % consistency | 55 | | | 29 | 51 | | | | | | | | | | | | |
| | # comparisons | 23 | | | 14 | 65 | | | | | | | | | | | | |
| | # points | 1361 | 0 | 0 | 1456 | 4948 | | | | | | | | | | | | |
| IAGOS China Airlines MSN-861 | start date | 2016/07/06 | 2016/09/05 | | | | | | | | | | | | | | | |
| | instr. serial | 14 | 16 | | | | | | | | | | | | | | | |
| | duration (days) | 61 | 449 | | | | | | | | | | | | | | | |
| | mean bias | 1.1 | 0.9 | | | | | | | | | | | | | | | |
| | % consistency | 50 | 56 | | | | | | | | | | | | | | | |
| | # comparisons | 37 | 28 | | | | | | | | | | | | | | | |
| | # points | 3698 | 1082 | | | | | | | | | | | | | | | |
| IAGOS Lufthansa MSN-989 | start date | 2015/03/08 | 2015/07/01 | 2016/01/13 | 2016/09/24 | 2016/11/19 | 2018/02/23 | 2018/06/16 | 2018/09/10 | 2018/12/12 | 2019/02/13 | 2019/08/16 | 2019/10/24 | 2020/02/13 | | | | |
| | instr. serial | 10 | 04 | 10 | 08 | 04 | 09 | 15 | 04 | 11 | 04 | 16 | 07 | 04 | | | | |
| | duration (days) | 115 | 196 | 255 | 56 | 459 | 113 | 86 | 93 | 63 | 184 | 69 | 112 | 77 | | | | |
| | mean bias | | -0.2 | 0.1 | | -0.1 | | 3.7 | -0.9 | -4.7 | 0.4 | -1.8 | | | | | | |
| | % consistency | | 49 | 44 | | 58 | | 22 | 84 | 66 | 57 | 66 | | | | | | |
| | # comparisons | | 8 | 33 | | 93 | | 1 | 1 | 2 | 21 | 1 | | | | | | |
| | # points | 0 | 376 | 2090 | 0 | 7070 | 0 | 98 | 136 | 233 | 2321 | 35 | 0 | 0 | | | | |

**Table 4.** Table compiling all the MOZAIC and IAGOS flight inter-comparisons $O_3$ measurement statistics and information for each instrument FPs. The latters are gathered and enumerated per aircraft. "Start date" is the date of the instrument unit installation on the aircraft (start of the flight period). "instr. serial" is the serial number of the instrument unit. "duration" is the lenght of the flight period in days. "mean bias" and "% consistency", see section 4.2 for explanation. "# comparisons" and "# points" are the number of collocated profiles and collocated data points found according to the methodology described in section 4.2.

**MOZAIC — Lufthansa — MSN-035**

| | FP n°1 | FP n°2 | FP n°3 | FP n°4 | FP n°5 | FP n°6 | FP n°7 | FP n°8 | FP n°9 | FP n°10 |
|---|---|---|---|---|---|---|---|---|---|---|
| start date: | 2002/06/04 | 2004/05/04 | 2006/02/01 | 2007/10/18 | 2008/03/07 | 2009/03/11 | 2010/11/24 | 2011/10/10 | 2012/11/13 | 2013/11/26 |
| instr. serial: | 02_PM | 06_PM | 01_PM | 02_PM | 03_PM | 04_PM | 03_PM | 02_PM | 01_PM | 02_PM |
| duration (days); | 700 | 638 | 624 | 172 | 338 | 319 | 624 | 400 | 378 | 98 |
| mean bias | -1.6 | 0.6 | 1.5 | -1.8 | 0.2 | -2.3 | -0.6 | -0.9 | 0.7 | -0.7 |
| % consistency; | 76 | 79 | 82 | 84 | 87 | 65 | 82 | 82 | 87 | 90 |
| # comparisons: | 42 | 131 | 169 | 57 | 21 | 107 | 67 | 142 | 164 | 28 |
| # points: | 3469 | 13673 | 20062 | 5163 | 1643 | 11518 | 6342 | 12326 | 13594 | 2384 |

**MOZAIC — Air France — MSN-049**

| | FP n°1 | FP n°2 | FP n°3 |
|---|---|---|---|
| start date: | 2001/04/30 | 2002/02/05 | 2003/05/21 |
| instr. serial: | 03_PM | 04_PM | 01_PM |
| duration (days); | 282 | 470 | 578 |
| mean bias | | 3.6 | 0.7 |
| % consistency; | | 67 | 88 |
| # comparisons: | | 4 | 30 |
| # points: | 0 | 153 | 3715 |

**MOZAIC — Multiple — MSN-051**

| | FP n°1 | FP n°2 | FP n°3 | FP n°4 |
|---|---|---|---|---|
| start date: | 2003/02/10 | 2003/04/23 | 2003/12/09 | 2004/11/22 |
| airline: | 01_PM SAB & DLH | 03_PM Lufthansa | 05_PM Lufthansa | 03_PM DLH & SW |
| duration (days); | 73 | 230 | 349 | 940 |
| mean bias | | -0.6 | -0.4 | -1.8 |
| % consistency; | | 78 | 90 | 78 |
| # comparisons: | | 49 | 142 | 96 |
| # points: | 0 | 4190 | 18780 | 10410 |

**MOZAIC — Lufthansa — MSN-053**

| | FP n°1 | FP n°2 | FP n°3 | FP n°4 | FP n°5 | FP n°6 | FP n°7 | FP n°8 | FP n°9 |
|---|---|---|---|---|---|---|---|---|---|
| start date: | 2001/11/27 | 2002/07/04 | 2002/09/17 | 2003/06/29 | 2005/04/21 | 2005/11/03 | 2006/10/29 | 2007/09/04 | 2008/05/08 |
| instr. serial: | 06_PM | 03_PM | 05_PM | 04_PM | 01_PM | 04_PM | 02_PM | 06_PM | 01_PM |
| duration (days); | 218 | 76 | 285 | 662 | 196 | 360 | 310 | 247 | 892 |
| mean bias | -3.3 | | 5.8 | -0.0 | 4.1 | -1.4 | -2.9 | 1.9 | 1.5 |
| % consistency; | 67 | | 68 | 87 | 79 | 83 | 83 | 84 | 67 |
| # comparisons: | 4 | | 20 | 212 | 79 | 148 | 71 | 84 | 139 |
| # points: | 149 | 0 | 2145 | 22715 | 7872 | 15009 | 9483 | 5026 | 14693 |

**MOZAIC — Austrian — MSN-075**

| | FP n°1 | FP n°2 | FP n°3 | FP n°4 | FP n°5 | FP n°6 | FP n°7 | FP n°8 | FP n°9 |
|---|---|---|---|---|---|---|---|---|---|
| start date: | 2002/02/04 | 2002/07/10 | 2002/12/09 | 2003/01/13 | 2004/03/04 | 2004/10/12 | 2005/12/02 | 2006/01/23 | 2006/05/12 |
| instr. serial: | 05_PM | 06_PM | 03_PM | 06_PM | 03_PM | 02_PM | 01_PM | 02_PM | 06_PM |
| duration (days); | 157 | 151 | 36 | 416 | 222 | 415 | 52 | 109 | 171 |
| mean bias | | -6.6 | | -0.8 | -2.7 | -0.1 | | | |
| % consistency; | | 63 | | 80 | 88 | 78 | | | |
| # comparisons: | | 1 | | 8 | 3 | 10 | | | |
| # points: | 0 | 272 | 0 | 789 | 526 | 799 | 0 | 0 | 0 |

**IAGOS — Iberia — MSN-221**

| | FP n°1 | FP n°2 |
|---|---|---|
| start date: | 2014/02/27 | 2016/02/17 |
| instr. serial: | 07 | 05 |
| duration (days); | 718 | 191 |
| mean bias | 3.2 | |
| % consistency; | 97 | |
| # comparisons: | 2 | |
| # points: | 186 | 0 |

**IAGOS — Lufthansa — MSN-304**

| | FP n°1 | FP n°2 | FP n°3 | FP n°4 | FP n°5 | FP n°6 | FP n°7 | FP n°8 | FP n°9 | FP n°10 | FP n°11 | FP n°12 | FP n°13 | FP n°14 | FP n°15 |
|---|---|---|---|---|---|---|---|---|---|---|---|---|---|---|---|
| start date: | 2011/07/07 | 2011/10/04 | 2011/12/12 | 2012/02/08 | 2012/04/04 | 2012/07/12 | 2012/10/29 | 2013/05/29 | 2013/12/08 | 2014/09/02 | 2015/03/30 | 2016/04/13 | 2016/07/08 | 2018/02/22 | 2019/05/18 |
| instr. serial: | 03 | 03 | 03 | 03 | 03 | 02 | 03 | 03 | 03 | 04 | 03 | 04 | 03 | 02 | 09 |
| duration (days); | 90 | 43 | 58 | 7 | 99 | 109 | 164 | 115 | 266 | 209 | 380 | 86 | 594 | 450 | 171 |
| mean bias | | 3.0 | 0.6 | 6.0 | -0.1 | -0.5 | -0.1 | -1.1 | 1.0 | | -0.4 | -2.7 | -2.9 | -1.6 | 0.9 |
| % consistency; | | 70 | 79 | 60 | 86 | 85 | 87 | 82 | 91 | | 91 | 81 | 77 | 87 | 91 |
| # comparisons: | | 6 | 21 | 3 | 43 | 45 | 79 | 40 | 27 | | 44 | 15 | 44 | 21 | 18 |
| # points: | 0 | 755 | 1822 | 248 | 4344 | 4301 | 7436 | 3008 | 2189 | 0 | 3409 | 689 | 3244 | 2055 | 1759 |

**IAGOS — Air France — MSN-377**

| | FP n°1 | FP n°2 |
|---|---|---|
| start date: | 2013/06/13 | 2016/04/19 |
| instr. serial: | 02 | 11 |
| duration (days); | 1040 | 609 |
| mean bias | 1.0 | 2.7 |
| % consistency; | 100 | 63 |
| # comparisons: | 1 | 11 |
| # points: | 67 | 974 |

**IAGOS — Cathay Pacific — MSN-421**

| | FP n°1 | FP n°4 |
|---|---|---|
| start date: | 2013/08/01 | 2016/11/17 |
| instr. serial: | 06 | 07 |
| duration (days); | 303 | 382 |
| mean bias | | -3.3 |
| % consistency; | | 62 |
| # comparisons: | | 18 |
| # points: | 0 | 965 |

**IAGOS — China Airlines — MSN-433**

| | FP n°1 | FP n°2 | FP n°3 | FP n°4 | FP n°5 |
|---|---|---|---|---|---|
| start date: | 2012/06/21 | 2014/06/18 | 2014/12/17 | 2015/07/23 | 2016/05/10 |
| instr. serial: | 04 | 08 | 09 | 08 | 09 |
| duration (days); | 485 | 182 | 218 | 292 | 388 |
| mean bias | -0.9 | | | 2.0 | 0.8 |
| % consistency; | 82 | | | 65 | 78 |
| # comparisons: | 40 | | | 27 | 25 |
| # points: | 3021 | 0 | 0 | 4291 | 1019 |

**IAGOS — China Airlines — MSN-861**

| | FP n°1 | FP n°2 |
|---|---|---|
| start date: | 2016/07/06 | 2016/09/05 |
| instr. serial: | 14 | 16 |
| duration (days); | 61 | 449 |
| mean bias | 0.4 | 1.6 |
| % consistency; | 90 | 63 |
| # comparisons: | 4 | 36 |
| # points: | 468 | 1474 |

**IAGOS — Lufthansa — MSN-989**

| | FP n°1 | FP n°2 | FP n°3 | FP n°7 | FP n°8 | FP n°9 | FP n°10 | FP n°11 | FP n°12 | FP n°13 |
|---|---|---|---|---|---|---|---|---|---|---|
| start date: | 2015/03/08 | 2015/07/01 | 2016/01/13 | 2018/06/16 | 2018/09/10 | 2018/12/12 | 2019/02/13 | 2019/08/16 | 2019/10/24 | 2020/02/13 |
| instr. serial: | 10 | 04 | 10 | 15 | 04 | 11 | 04 | 16 | 07 | 04 |
| duration (days); | 115 | 196 | 255 | 86 | 93 | 63 | 184 | 69 | 112 | 77 |
| mean bias | | 1.2 | 2.7 | 3.3 | 2.3 | | -0.5 | | | |
| % consistency; | | 91 | 86 | 72 | 98 | | 93 | | | |
| # comparisons: | | 14 | 68 | 7 | 3 | | 29 | | | |
| # points: | 0 | 707 | 4850 | 848 | 370 | 0 | 2596 | 0 | 0 | 0 |

**Table 5.** Same as Table 4 but for CO

Package MOZAIC and the IAGOS Package 1 (CARIBIC has its own deep blue round open circle). Figure 11 highlights that instrument mean bias greater than $\pm 2$ ppbv and $\pm 5$ ppbv for O3 and CO, respectively, are related to a low number of profiles that fit the criteria for the comparison (less than about 10) per FP. This is due to 3 main reasons: 1. if an airline has at least 2 of their aircraft equipped with the system, the number of flight coincidences might be reduced if one system isn't working properly for a long period of time. 2. if the equipped aircraft are located in different home hub. 3. if only one aircraft is equipped by that airline there will be fewer fight coincidences. The coincidences depend on the aircraft schedules which are not controlled by IAGOS. IAGOS therefore tries to equip more than one aircraft per airline. An other important result shown by Figure 11 is that the internal consistency of the MOZAIC and IAGOS instruments are similar. This result offers assurances that despite the differences in instrumentation since IAGOS began in 1994, the database of O3 and CO measurements can be considered homogeneous.

## 6 Conclusions

As pointed out in Tarasick et al. (2019), a lack of information on temporal changes in measurement uncertainties is an area of concern especially for long-term trend studies of the key compounds which have a direct or indirect impact on climate change. The IAGOS program (including MOZAIC and CARIBIC) has measured $O_3$ and $CO$ within the troposphere and the lower stratosphere for more than 25 years and represents the longest airborne time-series for these two gases with a large coverage in time and space, particularly in the northern hemisphere. Since 1994, the aircraft instrument setup has evolved to changing aeronautical regulations but much care was taken to keep the consistency of the measurement over time. This was achieved by using the same robust and well-recognized technologies based on UV absorption and IR correlation for $O_3$ and $CO$ and by following the the same calibration procedures from the beginning to now. In this study, thanks to many flight profile coincidences to compare the measurements made by different IAGOS aircraft, we demonstrated that the $O_3$ and $CO$ data, despite the change of instrument setup over time, present no drifts in bias over time. The study highlights the needs for the IAGOS program to increase the size of the fleet with at least 2 aircraft per airlines not only to increase the density of the measurement worldwide but also to be able to monitor closely the performances of each instrument unit mounted on-board.

*Data availability.* Boulanger, D., Thouret, V., Petzold, A. (2019). IAGOS Data Portal. Aeris. https://doi.org/10.25326/20

*Author contributions.* RB: investigation, methodology, formal analysis, visualization, writing-original draft preparation; PN: conceptualization, investigation, supervision; DB: data curation; PW: software; BS: conceptualization, supervision; JMC: resources; GA: software; AZ: project administration IAGOS-CARIBIC; FO: resources; DS: resources; HP: formal analysis; YB: software, formal analysis; HC: writing-review and editing; VT: funding acquisition, project administration IAGOS-CORE.

*Competing interests.* The authors declare that they have no conflict of interest.

*Acknowledgements.* IAGOS gratefully acknowledges financial support during its preparation, implementation, and operation phase from the European Commission in FP6 and FP7 programmes, the IAGOS for the GMES Atmospheric Service national (IGAS), research programmes in Germany (BMBF), France (INSU-CNRS, MESR, CNES), and the UK (NERC), in addition to institutional resources in Germany (Helmholtz Association, Max-Planck-Society, Leibniz Association), France (Université de Toulouse, Météo-France), and the UK (University of Manchester, University of Cambridge), and the continuing support of participating airlines (Lufthansa, Air-France, Iberia in Europe, China Airlines and Cathay Pacific in Asia). IAGOS wishes to emphasize the excellence of the industrial partners involved in the technical development: Sabena Technics Bordeaux for aircraft systems definition and certification, LGM Ingénierie for the instruments realization and aeronautic qualification, and Enviscope GmbH Frankfurt for operating the maintenance center of IAGOS.

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
