# Peer review of "Internal consistency of the IAGOS ozone and carbon monoxide measurements for the last 25 years."

_Atmospheric Measurement Techniques, 2020_

## Referee Comment (RC1) · Anonymous Referee #1 · 19 Feb 2021

Romain Blot et al. presents an interesting internal consistency study for the IAGOS ozone and carbon monoxide measurements based on the analysis of co-located take-off/landing profiles from different instruments of the network. Overall the paper is well written and is scientifically sound, my only concern is related with the filtering criteria and how they affect the results. I recommend it to be published after the following minor comments are addressed.

Specific comments:

The authors explain that the filtering criteria (wind direction, time difference, etc.) is a compromise between the similarity of the 'air masses' and the number of points. It

would be interesting to see what is the sensitivity of the analysis to some changes in these filtering parameters and how it affects the dispersion shown in, for example, Fig. 8. The ratio between the number of points within and outside the uncertainty as function of the change in the filtering criteria might help to understand what fraction of the observed measurements variability is actually due to differences in the sampled 'air masses'.

Fig. 7c shows some points above 9km which are quite far apart and show a large difference in ozone and carbon monoxide despite being considered as matching. Wouldn't it be good to add a distance criteria too? Would this dramatically affect the number of points?

Technical corrections:

When altitude is shown, please clarify if it is above sea level or above ground.

Fig. 7 looks messy. I would rearrange the panels/table to make it one figure and one table, and I would put all the captions in the figure caption instead of separate for each panel.

Fig. 9 and Fig. 10: You might want to reduce the size of the symbols (or make the legend larger) to avoid overlapping of the symbols.

L229-232: One sentence is repeated.

---

## Referee Comment (RC2) · Anonymous Referee #2 · 26 Feb 2021

Review of "Internal consistency of the IAGOS ozone and carbon monoxide measurements for the last 25 years" by Romain Blot et al., submitted to AMT

The paper presents good demonstration of an internal consistency of the IAGOS observation data since mid-1990s until now, for two chemical tracers - O3 and CO that are often used in the atmospheric chemistry research. The examined data are mainly from IAGOS-CORE and MOZAIC but also from CARIBIC as well.

The IAGOS data are often used in long-term trend analysis of tropospheric O3, and play a unique and critical role to fill the gap between ground-based and satellite observations. The authors tried to pick up co-located profiles, in both time and space,

and also put constraints on T, PV, and wind direction to assure that the instruments observed the same air masses, hence the differences in the observed mixing ratios can only be attributed to the instrumental differences. This is indeed a careful work, and they succeeded in demonstrating the stable operation of the entire observing system in such a long-term. The findings in this paper can act as a solid basis for the IAGOS data being used in long-term trends analysis of O3 and CO. Therefore, the paper is a great contribution not only to the data users but also to the wider atmospheric science community.

Overall, the paper is well written and nicely organized. In addition to the method, results and discussions, the paper includes a brief description of the IAGOS program, instrumentation, and standard operation procedure, which help the readers understand the basic components of the program. I have no major or minor critiques on the contents and agree that AMT is a good place for the authors to publish the paper as a good "interim (25 year's)" summary of the project to continue further.

Just a few technical comments are:

Page 13, Line 207: ...which fall within a maximum 1 hour time window... and other places. The authors say 1 hour as a time threshold. I guess this is <60 min, and if yes, perhaps good to clarify this somewhere. For example, the data pair of 01:10 vs. 02:50 is 1 hour difference, if we consider only "hour", but in fact the difference is 100 min.

Page 14, Line 221: even thought as... should be even though as... ? (typo)

Page 23, Line 331: space is needed before "IAGOS wishes..." (typo)

---

## Author Comment (AC1) · 24 Mar 2021

The authors thank the anonymous referee #2 for her/his constructive comments and corrections which have helped to improve our original manuscript. Referee comments which we are responding are given in small italics below.

**General comments:**

*The paper presents good demonstration of an internal consistency of the IAGOS observation data since mid-1990s until now, for two chemical tracers - O3 and CO that are often used in the atmospheric chemistry research. The examined data are mainly The IAGOS data are often used in long-term trend analysis of tropospheric O3, and play a unique and critical role to fill the gap between ground-based and satellite observations. The authors tried to pick up co-located profiles, in both time and space, and also put constraints on T, PV, and wind direction to assure that the instruments observed the same air masses, hence the differences in the observed mixing ratios can only be attributed to the instrumental differences. This is indeed a careful work, and they succeeded in demonstrating the stable operation of the entire observing system in such a long-term. The findings in this paper can act as a solid basis for the IAGOS data being used in long-term trends analysis of O3 and CO. Therefore, the paper is a great contribution not only to the data users but also to the wider atmospheric science community. Overall, the paper is well written and nicely organized. In addition to the method, results and discussions, the paper includes a brief description of the IAGOS program, instrumentation, and standard operation procedure, which help the readers understand the basic components of the program. I have no major or minor critiques on the contents and agree that AMT is a good place for the authors to publish the paper as a good "interim (25 year's)" summary of the project to continue further.*

**Reply:** Many thanks for the positive assessment concerning the general overview of our manuscript and the scientific outcomes of our study.

**Technical corrections:**

We thank the referee for all comments made below. All corrections will be done in the final manuscript.

*Page 13, Line 207:...which fall within a maximum 1 hour time window... and other places. The authors say 1 hour as a time threshold. I guess this is <60 min, and if yes, perhaps good to clarify this somewhere. For example, the data pair of 01:10 vs. 02:50 is 1 hour difference, if we consider only "hour", but in fact the difference is 100 min.*

**Reply:** We added to the text that, indeed, 1 hour as the maximum time difference between take offs or landings stands for a maximum of 60 min.

*Page 14, Line 221: even thought as... should be even though as... ? (typo)*

**Reply:** Corrected

*Page 23, Line 331: space is needed before "IAGOS wishes..." (typo)*

**Reply:** Corrected

---

## Author Comment (AC2) · 24 Mar 2021

The authors thank the anonymous referee #1 for his/her constructive comments and corrections which have helped to improve our original manuscript. Referee comments which we are responding are given in small italics below.

**General comments:**

*Romain Blot et al. presents an interesting internal consistency study for the IAGOS ozone and carbon monoxide measurements based on the analysis of co-located take-off/landing profiles from different instruments of the network. Overall the paper is well written and is scientifically sound, my only concern is related with the filtering criteria and how they affect the results. I recommend it to be published after the following minor comments are addressed.*

**Reply:** Many thanks for the positive assessment concerning the general overview of our manuscript.

**Specific comments:**

*The authors explain that the filtering criteria (wind direction, time difference, etc.) is a compromise between the similarity of the 'air masses' and the number of points. It would be interesting to see what is the sensitivity of the analysis to some changes in these filtering parameters and how it affects the dispersion shown in, for example, Fig. 8. The ratio between the number of points within and outside the uncertainty as function of the change in the filtering criteria might help to understand what fraction of the observed measurements variability is actually due to differences in the sampled 'air masses'.*

**Reply:** We wrote an additional paragraph and we added an additional Figure (Figure 9) to the section 4.2 (Methodology) in order to provide more details on how we choose the filtering criteria that we later use in the rest of the study:

"The air mass similarity criteria thresholds were found by testing the following method on several different FPs (flight periods). First, as it is shown in Figure 9 for the CO instrument $04_{PM}$ (same FP than in Figure 8), we monitor the evolution of performance indicators as we increase the temperature difference threshold. The percentage of points within the uncertainty peaks at a temperature difference of 0.25K and the mean bias peaks at 1K. For this flight period, we found 185 co-located aircraft which explain the large number (about 10000) of comparable points, even for a very restrictive threshold. Seeing the rapid increase of the number of points and comparing the results from others instrument units, we found that a temperature difference threshold of 1K would be a better compromise for shorter FPs or for the ones with instruments that are operated in remote area with less aircraft rotation (less co-located flights). To choose the two additional meteorological air mass similarity thresholds, we set the temperature thresholds to 1K and successively iterate on the wind direction and the potential vorticity difference increase for several FPs. Then, we decided to applied the same thresholds to all the FPs. For the time difference, we get better results for thresholds less than 1 hours (more steady meteorological conditions with respect to the life time of ozone and CO), however, we found that, for example at Frankfurt airport, the number of co-located flights is reduced by 50% per 30 minutes."

[Figure]

**Figure 9.** Percentage of points within the uncertainty and mean bias as function of the potential temperature difference measured by the co-located aircraft found for CO instrument $04_{PM}$ and for the flight period number 13 on Lufthansa A340msn53. The total number of comparable points according the air temperature difference is also shown. The maximum time difference between the landings or the take offs is 1 hour.

*Fig. 7c shows some points above 9km which are quite far apart and show a large difference in ozone and carbon monoxide despite being considered as matching.*

**Reply:** The grey lines mentioned above 9km that would suggest the air masses matching between the 2 flights are due to a coding minor error. There is a negligible impact on the scatter plots and the calculated statistics shown in Figure 7d. We carefully recalculated and updated all the statistic numbers in table 4 and 5 and the plots shown in Figure 8, 9 and 10 (Figure 8, 10 and 11 of the revised manuscript).

*Wouldn't it be good to add a distance criteria too? Would this dramatically affect the number of points?*

**Reply:** We decided not to add a distance criteria but only use the meteorological parameters. This is clearly the strictest criteria to check the air mass matching. In the free troposphere, thanks to their life time, ozone and CO may be very similar over large areas. Regarding the regional distributions that IAGOS provides, it is accepted that the vertical variability is higher than the horizontal one. Therefore, the thermodynamical parameters are better suited to confirm the regional horizontal homogeneity. A distance criterion would be (i) difficult to define non-arbitrarily, and (ii) it would cancel some possible comparisons. It clearly depends on the meteorological synoptic situation.
For example, the Figure 1 (see below) added to our answer, shows 3 different aircraft on their approach to Frankfurt on 2013/10/19. As you can see, the aircraft equipped with the CO instrument 04_PM (green line) arrived from the East whereas the other flights arrived from the West. Still, we found good air masses matching that shows very similar CO concentrations. A distance criterion would have ignored this type of interesting event, and lowered the resulting statistical robustness.

[Figure]

*Figure 1: Inter-comparison of 3 flights during their descent to Frankfurt on 2013/10/19.*

**Technical corrections:**

We thank the referee for all the comments made below. All the corrections will be done in the final manuscript.

*When altitude is shown, please clarify if it is above sea level or above ground.*

**Reply:** In this study, we refer to the aircraft barometric altitude that is derived from the aircraft altimeter. It is considered as the altitude above mean sea level. We added the clarification in the text and the figures.

*Fig. 7 looks messy. I would rearrange the panels/table to make it one figure and one table, and I would put all the captions in the figure caption instead of separate for each panel.*

**Reply:** Fig. 7 sub-figures arrangement is designed for the 2-columns .pdf final format of the AMT journal. We think that the lack of clarity mentioned is mainly due to the 1 page manuscript format asked for the peer review submission.

*Fig. 9 and Fig. 10: You might want to reduce the size of the symbols (or make the legend larger) to avoid overlapping of the symbols.*

**Reply:** We added larger inter-spaces between the labels in the legend and we reduced the size of the symbols to improve the clarity of the Figures.

*L229-232: One sentence is repeated.*

**Reply:** Duplicate sentence deleted